# Means of Livelihood, Clean Environment to Women Empowerment: The Multi-Faceted Role of Donkeys

**DOI:** 10.3390/ani13121927

**Published:** 2023-06-09

**Authors:** Thanammal Ravichandran, Ramesh Kumar Perumal, Kennady Vijayalakshmy, Zoe Raw, Fiona Cooke, Isabelle Baltenweck, Habibar Rahman

**Affiliations:** 1International Livestock Research Institute (ILRI), New Delhi 100012, India; 2Kumaraguru College of Liberal Arts and Science, Coimbatore 641049, India; 3Research, Research and Operational Support, The Donkey Sanctuary, Sidmouth EX10 0NU, UK; 4Research & Ecology, Research and Operational Support, The Donkey Sanctuary, Sidmouth EX10 0NU, UK; 5Policies, Institutions and Livelihoods, International Livestock Research Institute (ILRI), Nairobi P.O. Box 40241-00100, Kenya; 6Regional Representative for South Asia, International Livestock Research Institute (ILRI), New Delhi 100012, India

**Keywords:** donkey welfare, population decline, sustainable development goals (SDGs), working equids, human livelihoods

## Abstract

**Simple Summary:**

Donkeys are resilient species and exhibit functional adaptation in a wide range of climatic conditions. However, in comparison to other livestock species, their inclusion in agricultural or livestock health and nutrition policy, education, and research sectors is negligible. The focus of the current review is on the value of donkeys across a range of sectors, including agriculture, construction, industry, and tourism. The study emphasizes the importance of donkey welfare, as well as the vital function that donkeys serve in sustaining the livelihoods of some of the most underprivileged communities all around the world. In poor and oppressed communities all around the world, donkeys can help to empower women and give them a way to work and make a living.

**Abstract:**

Despite the substantial contribution donkeys make to the livelihood of the world’s poorest populations, the existence of donkeys has received little notice worldwide. This article reviews the value of donkeys in a variety of sectors, including agriculture, construction industry, and mining, as well as their role in empowering women and achieving sustainable development goals. However, donkeys and mules are not given enough credit or attention in terms of developing strategies regarding their role in reducing poverty. There is a dearth of information and statistics on their impact across industries, the factors contributing to the donkey population dropping, the socioeconomic status of the dependent communities, and related animal and human welfare issues.

## 1. Introduction

Donkeys and mules are non-ruminant members of the Equidae family found in a range of ecological zones, including semi-arid, temperate, and highlands, across the globe. Mules are the offspring of a male donkey (Jack) and a female horse (mare), while hinnies are the offspring of a female donkey (Jenny) and a male horse (stallion). Throughout this manuscript, the word mule is applied to refer to both mules and hinnies. The domestication of donkeys dates back to over 7000 years ago [1,2,3] when they played an important role as pack or draught animals [4]. For the last three decades, there has been a continued increase in the global donkey and mule population, especially in Africa, South Asia, and Latin America [5]. They are a resilient species, capable of covering long distances whilst remaining thirst tolerant, and are able to rehydrate quickly with intermittent access to water (whereas other pack animals, i.e., horses, need to drink more often). Because of this, donkeys were domesticated as a means of transportation over longer distances [6]. Donkeys can also thrive on low-energy-density feedstuffs, although this does not preclude the need for adequate nutrition [7].

Developing countries are home to 96% of donkeys and 60% of horses globally [8,9]. In rural and peri-urban areas, working animals—particularly equids—are a primary source of income and sustain the livelihood of large families. Each working equid may support up to 20 people to fulfill their daily survival needs [10]. Donkeys are a vital component of rural development and a significant mode of transportation for pastoral communities. They enable farmers to access distant markets in which to sell their produce, which would be inaccessible if it were not for their donkeys being able to carry their products over long distances [8]. Donkey traction energy is a cost-effective alternative to human traction or tiny fossil fuel-powered vehicles in the construction industry. For instance, donkeys and mules can navigate the uneven terrain of mountainous areas and brick kiln sites with greater efficiency than most mechanized vehicles [11].

Despite the important role played by donkeys in the socio-economic fabric of rural and peri-urban areas, not much importance has been accorded to their existence in most parts of the world [8]. The welfare of donkeys is a major concern, especially given that they are typically owned by resource-poor owners and are kept in challenging environmental conditions, such as mountainous, arid, or semi-arid regions, often with little access to veterinary or healthcare services [8]. This review article aimed to explore the population trend of donkeys worldwide; the declining trend, specifically in India; and their importance in draught power, clean environment, livelihood support, and women’s empowerment.

## 2. Donkey Population Trend: A World in General and Specific in India

According to the study by [12], the accuracy of the estimated donkey population could be contentious, although the data on donkey population are gathered and maintained by respective agricultural ministries nationally. Based on this information or through linear approximation, global numbers are interpolated by the United Nations Food and Agricultural Organization (FAO). This is because these working animals are mostly found in rural remote areas where they are either impossible to survey or for which there is a lack of records of ownership of these animals [4] (Fielding and Starkey, 2004). Based on the donkey population data available on the FAO website, it can be concluded that the world witnessed an increase in its population starting in 1961. According to the United Nations Food and Agricultural Organization (FAO), population estimates of donkeys around the world have increased from 38 million in 1968 to 50 million in 2018 [13]. While the global donkey population has shown an upward trend, there are large regional differences; for example, most of the donkeys were found in Sub-Saharan Africa, the Northern region of the Indian Subcontinent, and Latin America [5].

Although the estimated population of donkeys in developing countries has risen for the last four decades, their population has decreased significantly in India. According to the 20th livestock census, the donkey population has declined from 0.32 million in 2012 to 0.12 million in 2019—a decline of 61.23% (Table 1). Over a period of 27 years, India has registered an almost 88% decline in donkey population from 967,000 in 1992 to 120,000 in 2019. While a steep decline has been observed in the donkey population, the horse population also registered a gradual decline over the years, unlike mules whose population was observed to show a rising trend till 2012 and declined in 2019 (Figure 1). This increase in the mule population is especially noticeable in northern parts of India, where mule traction is often used in the brick kiln industry and for tourism purposes at pilgrim sites. Another important reason for a sharp reduction in the donkey population could be outbreaks of glanders during the last 10 years, as donkeys get the acute form [14].

A rapid decline in local donkey populations is also being observed in some (although not all) African countries. The decline in African donkey populations is thought to be related to the increased export of donkey hides to supply the expanding demand for traditional Chinese medicine, ‘ejiao‘. According to [15], the increased exports are hampering the livelihoods of South African donkey-owning communities for whom donkeys are the primary source of income.

## 3. Domestication of Donkeys and Their Role in the Ancient Period

The domestication of the African wild ass revolutionized ancient transit methods, as well as the organization of early cities and pastoral cultures in Africa and Asia. Donkeys are uncommon in the archaeological record, and evidence for early phases of animal domestication is difficult to detect; therefore, genetic research implies an African origin for the donkey. However, pinpointing the period and location of domestication has been difficult because donkeys are uncommon in the archaeological record [6]. According to this literature, donkeys were used mostly for traction power rather than for meat.

There is a powerful contention that it was pastoralist cattle herders in northeastern Africa who originally enlisted local jack asses to work in their herds in the fifth millennium BC, for shipping their camp belongings, and to fetch water from far-off water sources [16]. The contribution of donkeys to global trade and commerce started as early as the second half of the 15th century. According to the research by [17], the establishment of important trade routes coincides with the spread of donkeys throughout the Caribbean and Latin American regions. Historical evidence from rock art or ancient scriptures also emphasizes the important role of donkeys during warfare. They are depicted as being used to transport supplies for soldiers and for carrying sick or wounded personnel for medical treatment [7].

According to ancient literature, donkey milk was used by Hippocrates, the father of modern medicine, who used it to treat liver problems and infectious diseases. In particular, donkey milk is regarded as a good source of nutrition for treating conditions such as psoriasis and dry skin that are related to the skin; joint pains, asthma, stomach ulcer, sore throat, liver problems, infectious diseases, fever, edema, poisoning, and wounds [18,19,20,21,22].

## 4. Draught Power of Donkeys

Donkeys are considered an important source of draught animal power in arid and semi-arid regions, as they can tolerate drought conditions more successfully than oxen, and they can tolerate infrequent access to water [23]. Many reports stated that there are an estimated 112 million working equids in the world which support the lives of approximately 600 million people [24]. There is evidence that the draught power of just one equid can be the only source of livelihood for an entire family unit, making human and animal welfare inseparable [25]. According to the research by [26], draught power has always been considered a non-marketable service and has not been adequately accounted for, as an important contribution of livestock GDP in agriculture. He emphasizes that although traction energy may be considered a less tangible contribution, it makes an important contribution to accrued economic profits. Adaptation to mountainous environments allows donkeys to easily tread over steep, rocky, and narrow landscapes, and both male and female donkeys can be effectively used for work [27].

Donkey draught power has had a significant impact on highland communities, decreasing their drudgery. A study by [28] reported that post-harvest food losses were incurred high in Uganda’s highlands due to a lack of farm transport. The introduction of donkey transport to these areas had a multi-faceted positive effect, not only providing affordable and sustainable transport technology but also reducing the drudgery in day-to-day lives [28]. This is applicable specifically to women, as they are the ones who are associated with harvest and post-harvest management-related farm activities. There is a lack of recent data on the economic contribution of traction power by draught animals in developing countries. However, according to older estimates, working animals contributed almost 75% of draught power in developing countries [29]. This has undoubtedly been reduced in the last two decades due to mechanization in all sectors which replaced animal draught power. The income from equine sources has been neglected due to their indirect contribution to revenue generation from agriculture and livestock products sale. Equines were considered an important draught power when transporting agricultural products to market for sale. These households were unable to separate equine-related costs from livestock-related costs, making it impossible for them to estimate such costs [10,30].

## 5. Cultural Values and Other Influencers Associated with Equid Ownership

Different communities within the population own donkeys in India. A recent study by [31], points out that donkey ownership in northern parts of India is mostly dictated by caste and ethnicity, which also happens to be linked to impoverished families with low status and limited access to resources. The author asserts that such systems are parallelly oppressive, both to the people and to the animals managed by these communities. For example, in the brick kiln industries, both humans and their donkeys are exploited, experience poor welfare and undergo a violation of their basic rights.

Affordability is a key factor influencing the choice of working equids made by the potential owner or trader. Donkeys are a more economical option compared to mules and horses which are expensive to purchase and maintain; in India, the price of a donkey may vary between USD 70 and 175 on average compared to the price of horses and mules, which ranges between USD 600 and 1000 on average (Source: FGD with the equid owner in Maharashtra by the first author). The price of an equid is based on its health, age, and body condition; for example, equids in poor body condition fetch a lower price. Other factors which may influence the purchase price are the competency of the buyers in equid selection, beliefs and perceptions, requirements of working capacity, maintenance capability, and owners’ preferences [11]. Most of the donkeys in India are owned by landless communities. Certain communities such as Kumhars (potters) and nomads prefer only mares, as they feel they are docile and easy to manage, whereas some (Vanjara) prefer only stallions. The belief among donkey owners is that stallions have more capacity to work (especially in brick kilns) than geldings and mares. Additionally, donkeys are believed to be sturdy and more resilient to drought than horses and are perceived to be easily managed by children or women. They are mostly controlled using sticks and whips, since their behavioral responses to fear and pain are less overt than in horses [32], although this does not reduce the need for correct and humane handling or management.

## 6. Importance of Donkeys for Livelihood of Smallholder Farmers and Transporters

In rural and peri-urban areas, working animals—particularly equids—could be the primary source of income and livelihood, sustaining the family units of hundreds of millions of impoverished people. Each equid supports up to 20 persons to fulfill their daily survival needs [10]. The participation of small-scale farmers in the rural market economy is enabled through the traction power supplied by draught animals that ensure the timely movement of produce from fields to markets, allowing easy connectivity with traders, and improving market access that translates to increased profits [30,33]. Good husbandry and healthcare services for draught animals go beyond animal welfare and are a means to strengthen the livelihoods of their owners [34].

To mobilize and enable active support from a variety of sources, such as government or private extension workers, veterinarians, community-based animal health workers, and field staff from community organizations, collective action is important for the poorest communities that raise draught animals. Through participatory action research, development actors (NGOs/extension agents) and farmers who rear draught animals can learn more about equine welfare and find suitable solutions. On a related note, there is an exemplary case study on equid welfare in Uttar Pradesh, a northern state in India where horse owners worked collectively to identify the cause of persistent wither lesions in their equines. While the risk analysis observations were facilitated by the equine welfare organization, the horse owners solved the issue as a group and arrived at a local solution that was effectively implemented. It is equally important to provide a similar case in examples of the knowledge of policymakers to highlight the importance of donkey welfare as a contribution to strengthening the livelihood of their owners. This will facilitate the devising of policies that aim to achieve the improved welfare of working animals in the longer term [8].

## 7. Contribution of Donkeys toward Achieving Sustainable Development Goals (SDGs)

The Sustainable Development Goals (SDGs) are set by the United Nations (UN) and designed to create a more sustainable future for humanity. According to a recent report by the charity Brooke, donkeys play an important role through direct and indirect support in income-generating activities in a variety of sectors such as agriculture, transport, tourism, and the construction industry; they play a vital role in poverty alleviation, as described in SDG1 [16]. A study by [11] on the construction industry in India reveals that working equids contribute to the generation of 80% of income of the employed workforce. The income generated from the utilization of the traction energy of donkeys secures the buying capacity of their owners and rearers to afford food and feed for their families and livestock. Donkeys enable access to local food markets for nutrition and distantly located water resources contributing towards SDG 2: Zero hunger [35]. Donkeys, through their day-to-day activities, contribute towards reducing drudgery for women, fetching water and firewood from long distances and transporting farm produce to fetch income for women-headed households, and thus contributing to SGD 5 of Gender equality [36]. Donkeys help to build resilience; the extra income they generate allows people to save money, reinvest in growth, and access education [37], which explains their contribution to SDG 8: Decent work and economic growth. Access to clean drinking water in rural remote areas requires the time and labor of millions of people across the world. Draught power from donkeys is readily used by rural households to enable access to water for families and livestock (SDG 6: Clean water and sanitation) [35]. Donkeys adapt well to life in high temperatures in arid and semi-arid regions. They act as a buffer against disasters and climate shocks caused due to climate variations and contribute towards SGD 13 on climate action through rebuilding infrastructure or relocating families through transportation via clean energy. The human labor, animal welfare, and environmental sectors must work together more proactively to find integrated solutions, building on key linkages such as health, and existing partnerships such as the Clean Air and Climate Coalition’s Brick Production initiative and emerging framework of “One Health –One Welfare” [11,15].

## 8. Role of Donkeys in Adaptation and Mitigation of Climate Change

Strong evidence of working equids’ importance in assisting and supporting the resilience of communities in Low- and Middle-Income Countries (LMICs) will be necessary to persuade policymakers to recognize them (and provide the budgets to support them) in difficult and complex emergencies, especially given the emergence of climate-related crises. Researchers and practitioners will have to provide proof of working equid contributions in these at-risk areas, which is currently lacking but is essential to improving working equids’ standing in international policy and humanitarian agendas. This evidence might be concealed; for instance, if there are other, more important tasks to attend to when coping with a crisis. As a result, records documenting the contributions of working equids might be lost or forgotten [38].

With the increase in events of high temperatures and growing seasonality that affect rainfall patterns, leading to frequent floods or cyclones, the contribution of animal-powered transport technology should be both acknowledged and valued. Donkeys are drought tolerant and can traverse and transport over long distances [8]. Animal traction energy supersedes other polluting transportation that is reliant on non-renewable fuel sources.

## 9. Donkeys Support the Livelihoods of Brick Kiln and Mine Workers

It is a common practice to use donkeys, mules, and horses in the construction industry in the South Asian region. They are known to play an especially important role in brick production industries in India, Pakistan, Nepal, and Afghanistan [10,31]. Their traction energy is an economical replacement for fossil-fuel-driven small vehicles or physically intensive human traction. In addition, brick kiln owners prefer equids for the transport of bricks to motorized vehicles, as there is less breakage of green bricks which are kept in the kiln for burning. (This was determined through author interaction with brick kiln owners in India). The uneven topography of brick kiln sites can be efficiently trodden by donkeys and mules [11]. Approximately 380,000 equids support the work of brick loaders in the Indian brick kilns, where they transport molded bricks into the kilns. Donkeys are mostly used in mines for the transport of coal and materials, as they are believed to be very sturdy. There is a lack of information and literature on the evaluation of the economic contribution of draught animals to industrial development.

In a survey conducted by The Brooke on the brick kiln industry in 10 districts of the state of Uttar Pradesh in 2013, of the total annual income earned by equid-owning families, 80% was generated from brick transportation by the animals. During the brick kiln season, 23.5% of equid owners depended entirely upon the services of their animals as their only source of income [39].

## 10. Working Equids and Their Role in Women’s Empowerment

Donkeys in particular can play a key role in the empowerment of women in developing countries by generating income and influencing social status [12,40], though both donkeys and women in many circumstances suffer a marginalized social status compared to men. This is especially true for working donkeys, who are frequently thought to be inferior to ruminant livestock in terms of value. Because of this, donkeys are frequently underestimated for their economic and social benefits to these populations, which causes government policymakers to neglect them when developing projects [37]. In many countries, handling an ox or other large livestock is traditionally considered a male activity due to their size and link to status (both social and financial), and so women are denied access [41]. Donkeys are regarded as women’s animals by being cheaper to purchase, accessible and acceptable for use [41], and potentially easier to handle and manage. In terms of attitude, donkeys are intimately linked to poverty, and utilizing donkeys for tasks often carries societal shame. The way people feel about donkeys is evolving, and they are gaining more “social worth.” Despite some restrictions brought on by cultural perceptions and technological advancements related to donkeys, women have demonstrated that they may particularly benefit from employing donkeys for both domestic and income-generating tasks. Compared to larger livestock, donkeys alleviate the drudgery undergone by women in rural areas regarding the transport of agricultural produce from farms to warehouses near homesteads. This is in addition to the traditional, arduous, and time-consuming chores such as fetching fuel wood or water from distant places, tasks which are undertaken exclusively by rural women [27]. The spared drudgery and time from using equids enables women to participate in social or community activities, earn an additional income, or gain some much-needed rest [42]; for example, according to a study conducted by [12], in a comparison of the daily routine of two Maasai women it was found that the one using donkeys for household chores saved almost 25 h a week compared to the one who managed all the activities herself. Women rely on donkeys to carry out tasks they would otherwise have to perform themselves, from collecting water, tilling the land, and transporting goods which enables them to be economically active and increase their community status and personal resilience. This economic capability can prevent the worst form of destitution for lone women [37].

## 11. Loss of Genetic Biodiversity of Donkeys

According to the research by [43], the decreasing donkey population across various regions in Italy and India is a serious cause for concern regarding the loss of genetic diversity. This outcome is thought to be the result of a lack of policy measures to conserve donkey breeds. Little information is available regarding different donkey breeds and their phenotypic characteristics in India [44]. Recent studies were carried out to characterize some donkey breeds in India, including Spiti donkeys in Himachal Pradesh [45] and genetic evaluation of donkey breeds from different agro-climatic zones [43].

Loss of genetic diversity can have substantial consequences when animals have evolved and adapted to geographical locations through centuries of natural selection; if these animals face changes in their environment, they are less able to adapt, thrive, and evolve in response [46]. Since genetic variability plays a key role in the future development of livestock, losses in genetic diversity may be an impediment to maintaining healthy donkeys that experience good welfare under a variety of conditions. Donkeys are also known to play an important role in conserving floral and faunal biodiversity. In areas where they are non-native, donkeys are considered invasive because they have the potential to endanger native species and natural environments. This has important implications for introduced free-roaming donkeys, which are frequently classified as non-native unnatural animals and treated as such. Modern conservation efforts focus on a variety of objectives, such as maximizing biodiversity, safeguarding a specific species or assemblage, preserving or restoring ecosystem services, or re-establishing a former biological community [38]. Research shows that the selective feeding ability of donkeys encourages the conservation of certain plant and insect species which have declined due to intensive agricultural practices. This may be an important contribution to the maintenance of scrubby meadows and grazing lands [47]. According to the study by [48], the donkey population in Europe has been declining in the last century due to mechanization. However, due to the fragility of low mountainous regions in some parts of Italy donkeys are preferred, over bulky farm animals such as cattle, for the control of grass growth [43,45].

## 12. Donkeys and Their Associated Health and Welfare Problems

The traction energy of equids, which are frequently seen in rural, isolated parts of developing countries, is most effectively used by poor communities. These landscapes and the weather they are linked with can be difficult, requiring draught animals to work tirelessly for lengthy periods of time. Further, these animals may suffer from a lack of appropriate nutrition due to limited access to clean drinking water and feed due to the financial hardship of their owners and their lack of land ownership. This situation is particularly challenging during dry seasons. Donkeys with poor Body Condition Scores (BCS) are prone to dehydration when they are working in harsh conditions [49]. Poor BCS are typically due to donkey owners not being able to afford adequate feed to meet the nutritional needs of their donkeys, and subsequently many are allowed to roam free and forage for themselves. Little or no availability of healthcare services in remote areas tends to aggravate debilitating health conditions, leaving the animals to suffer from an increased parasitic load, wounds, and injuries to skin and hooves, leading to pain and lameness, respiratory and gastrointestinal problems, and poor-condition animals [8,50,51]. In northern India, it is common practice to sell antibiotics to equine owners without a prescription and, most of the time, with the wrong diagnosis, course of treatment, and dosage. Few employees at Drug Retail Outlets (DROs) are adequately trained or certified to dispense antibiotics to animal owners, leaving a significant gap in their abilities. The study emphasizes the need for more in-depth training for private DRO employees, as well as education about antimicrobial resistance and its possible effects on livelihoods for both DRO employees and animal owners. It also demonstrates the need to strike a balance between implementing stricter control at all levels and ensuring rural populations have adequate access to medicine [52].

A study conducted by [8] to perform a welfare assessment of working equids across five developing countries, including Afghanistan, Egypt, India, Jordan, and Pakistan, reported an increased incidence of gait and eye abnormalities and lesions of the hind-quarter region in donkeys compared to horses or mules. The study attributed these problems to persistent beating or injuries borne from the cart or pack saddle. In this study, donkeys from Pakistan also demonstrated chronic disease conditions indicated by a drop in the hemoglobin count. According to [53], the health and welfare problems of donkeys may differ also based on their work locations, be they rural or urban or prevailing environmental conditions. For example, in this study across nine developing countries, it was observed that donkeys in rural areas were more prone to ectoparasitic infections, tendon or joint swellings, gait abnormality, and a lower body condition score. Comparatively, the donkeys in urban areas suffered less with sole surface problems due to tarmac roads. BCS was observed to be negatively correlated with temperature, while humid conditions were linked to gait abnormalities and wet seasons with swellings in tendons and joints. The aforementioned study also established that poor BCS of donkeys could be a good indicator of welfare problems such as lesions. Parasitic load and abnormalities of gait and sole were also observed more frequently in donkeys with a lower BCS. Since the welfare problems of donkeys differ demographically, BCS could be an important criterion to assess and address this problem [53].

## 13. Policy and Advocacy Efforts for Donkey Welfare

According to the study conducted by [8], policymakers both domestically and globally undervalue the socio-economic importance of draught animals to the rural development sector. As a result, these animals are ignored in policies and developmental schemes and appropriate technologies are not promoted to improve their welfare. Another barrier preventing the expansion of support for the well-being of draught animals is the absence of infrastructure to provide preventive and curative veterinary services. Public services and infrastructures are more inclined towards food animals than draught animals. Lately, there have been several efforts and initiatives to train and depute community animal health workers to provide para-veterinary healthcare services [54], although they have remained limited to food-producing livestock. It is, however, most imperative to create and fulfill the demand for veterinary health care services for working equids.

In a recent policy development work, global standards for the welfare of working equids have been adopted by the World Organization for Animal Health (OIE). The first of its kind includes guidelines on the living and working conditions of the donkeys, mules, or horses, used for or retired from traction, transport, and generation of income. These standards offer a basis for the formulation of policies and regulatory instruments to promote equid welfare in various sectors, including the construction industry. It is required of the OIE members of all countries to take requisite actions to implement these standards [55].

As part of the animal welfare indicator (AWIN) project of the European Union, the AWIN welfare assessment protocol for donkeys has been developed with the purpose of disseminating information about their welfare indicators. The protocol is based on four welfare quality principles, namely good feeding, good housing, appropriate behavior, and good health [56].

Numerous difficulties must be overcome to measure animal well-being, but one promising strategy is to combine existing knowledge into fresh frameworks that can provide uniform methods for doing so in a range of situations. The welfare of equids can be measured using a variety of measures; however, these tools are frequently created for specific circumstances. There is no “one size fits all,” and therefore the resulting datasets are typically incomparable, which makes it difficult for the various organizations attempting to improve equine welfare around the world to share information and work together. To address this, The Donkey Sanctuary (TDS) team created the Equid Assessment, Research and Scoping (EARS) tool, which combines already validated welfare assessment techniques with new welfare indicators to deliver a larger and more comprehensive set of welfare indicators than those that are already available [57]. This results in a single resource that can be used to evaluate equid welfare in any context. Three welfare assessment methods were field tested by the Donkey Sanctuary (TDS) team using the EARS tool to measure the welfare of equids in a range of settings across 19 countries.

## 14. Emerging Markets to Promote Donkey Conservation in Developed Countries

Several initiatives have emerged that promote sustainable donkey production to harness emerging markets and promote donkey conservation. There has been an increase in the demand for some donkey products. For example, donkey milk is sold at a premium price in some cities due to its alleged therapeutic and extra nutritional benefits. The application of donkey milk is especially being researched for consumption by infants [58]. Some of the purported benefits are reported to offer cures for skin diseases such as psoriasis; relief from joint pains and breathing problems such as asthma; and dental problems. It also has supposed cosmetic applications, such as anti-aging properties. This has resulted in the development of farms to rear dairy donkeys across the globe.

The composition of donkey milk is found to be closer to that of human breast milk regarding its lactose content and protein constitution [59]. It also has a higher concentration of vitamins and macro- and micro-minerals than other types of milk [60]. It has been clinically tested on children and is known to meet their nutritional requirements while having good palatability [17].

An increase in the consumption demand for donkey meat and milk has been observed in developing countries. This is being viewed as an incentive towards investing for improvement in donkey husbandry and production [7]. Brazil is witnessing the emergence of new companies that are investing in sustainable donkey production businesses [17]. Serious animal welfare problems can occur if appropriate measures are not implemented for the well-being of animals for such activities [61]. Donkey milk is sold in some parts of India, particularly the southern region, for infant nutrition and therapeutic purposes [62]. However, the consumption of donkey milk is not legalized and it is not certified to be consumed as milk. Making cosmetic products from donkey milk can enhance the donkey milk value chain. Communities in India that own donkeys, however, are unaware of these advantages, since they cannot afford proper feeding procedures and sourcing for milk products.

## 15. Conclusions

It is evident from the literature that working equines contribute much to the sustainable development goals by supporting the livelihood of the poorest families worldwide. They are considered sources of employment in various sectors, including agriculture, construction, tourism, and the mining sector. However, their contribution to enhancing the livelihood of the poor and welfare issues, especially in the case of the donkeys and mules, is underacknowledged and neglected in policies and development programs due to the lack of information and data to support their contribution. Efforts by various animal welfare organizations to improve the welfare of working equines have not achieved significant positive changes. When the health and welfare of working equines are not addressed in policies, it leads to poor welfare of equines, which subsequently has a negative impact on the income of communities dependent on them. This will lead to a trap of poor welfare of animals as well as humans.

There is a need for one welfare approach where the welfare of animals and humans are considered interlinked to each other, so a change in human welfare will bring positive change in animal welfare and improved animal welfare will increase productivity and household income. There are no data or information available at the national level to highlight the welfare issues of the donkey and mule population and their dependents to include them at the policy level. There is a need for detailed mapping of the donkey and mule populations in developing countries, the socio-economic status of communities who own them, their usage pattern, and the welfare issues of both humans and animals for a better understanding of sustainable intervention planning. There is a need for research and evidence to prove that improved human welfare increases the animal welfare condition and vice versa. Further study is needed to explore the emerging market for donkey milk, human nutrition, and its implications for the poorest communities.

## Figures and Tables

**Figure 1 animals-13-01927-f001:**
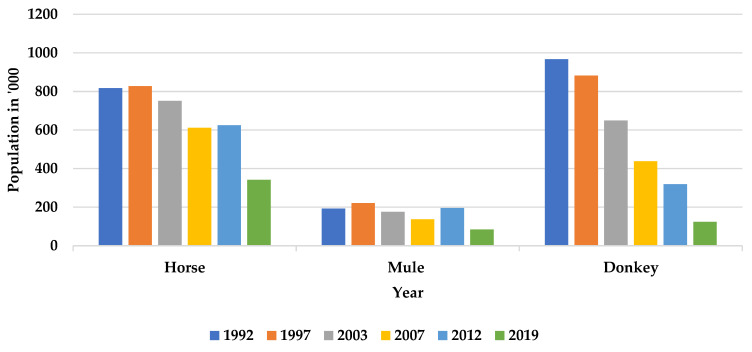
Equine population trend in India from 1992–2019 (Source: DAHD, 2022. Department of Animal Husbandry and Dairying, Ministry of Fisheries, Animal Husbandry and Dairying Government of India—Annual Report 2022–2023. https://dahd.nic.in/sites/default/filess/FINALREPORT2023ENGLISH.pdf, accessed on 18 November 2022).

**Table 1 animals-13-01927-t001:** Trends of Donkey Population in major states of India between 2012 and 2019 (Source: DAHD, 2022. Department of Animal Husbandry and Dairying, Ministry of Fisheries, Animal Husbandry and Dairying Government of India—Annual Report 2022–2023. https://dahd.nic.in/sites/default/filess/FINALREPORT2023ENGLISH.pdf, accessed on 18 November 2022).

S.No.	States	Population (In Lakhs) 2012	Population (In Lakhs) 2019	% Change
1.	Rajasthan	0.81	0.23	−71.31%
2.	Maharashtra	0.29	0.18	−39.69
3.	Uttar Pradesh	0.57	0.16	−71.72
4.	Gujarat	0.39	0.11	−70.94
5.	Bihar	0.21	0.11	−47.31
6.	Jammu and Kashmir	0.17	0.10	−44.55
7.	Karnataka	0.16	0.09	−46.11
8.	Madhya Pradesh	0.15	0.08	−45.46
9.	Himachal Pradesh	0.07	0.05	−34.73
10.	Andhra Pradesh	0.13	0.05	−65.16

## Data Availability

Not applicable.

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
