# Peer review of "Means of Livelihood, Clean Environment to Women Empowerment: The Multi-Faceted Role of Donkeys"

_animals, 2023, doi:10.3390/ani13121927_

Round 1

Reviewer 1 Report

The changes in the paper that I would recommend are to change the title of the paper and to indicate source for the Table 1 and Figures 1 and 2.

Author Response

Reviewer comments

Author responses

The changes in the paper that I would recommend are to change the title of the paper

Thanks for your suggestion Changed title of the paper “Means of livelihood, clean environment to women empowerment: the multi-faceted role of donkeys”

and to indicate source for the Table 1 and Figures 1 and 2.

Done

Reviewer 2 Report

Before publishing, some content and structural revision needs to be done. First of all, by doing a review it is important to underline the state of art analyzed and the gaps actually existing about the topic.

Abstract is poor in content and needs to be radically re-arranged in order to fit the aim of the review.   - lines 55 - 66 need to be written in a more simple and comprehensive way.   - lines 206- 211 should be rewritten. What is the connection with farmers?   - figure 1 and figure 2 need more description.   - paragraph 2 needs to be improved by more content and articles for the description.   - A material and method paragraph should be added.   The second part of the review needs to be cardinally arranged in a systematic way: the relation of workers, women and donkeys is not so clearly defined.   Paragraph 9 and paragraph 4 can be merged and connected.   Welfare issues may be improved.   A discussion paragraph is mandatory to evidence the knowledge and gaps.   Conclusion needs improvement.

Author Response

Reviewer comments

Author responses

Before publishing, some content and structural revision needs to be done. First of all, by doing a review it is important to underline the state of art analyzed and the gaps actually existing about the topic.

Thanks for your comment. The article revised for its content and structure. Research gap identified and reported in conclusion section

Abstract is poor in content and needs to be radically re-arranged in order to fit the aim of the review.

Thanks. Rewritten the abstract

lines 55 - 66 need to be written in a more simple and comprehensive way.  

done

lines 206- 211 should be rewritten. What is the connection with farmers?

done

figure 1 and figure 2 need more description.  

done

paragraph 2 needs to be improved by more content and articles for the description

done

A material and method paragraph should be added.  

This is a review article so material and method section not included.

The second part of the review needs to be cardinally arranged in a systematic way: the relation of workers, women and donkeys is not so clearly defined

Thanks. The article is reorganized for better flow of connection between donkeys, workers and women

Paragraph 9 and paragraph 4 can be merged and connected.   Welfare issues may be improved.  

A discussion paragraph is mandatory to evidence the knowledge and gaps.  

The article has been written with discussion in each topic. Conclusion section included the discussion and knowledge gap

Conclusion needs improvement

Thanks, and corrected

Reviewer 3 Report

The article is interesting and presents a general overview of the donkey supply chain in the relevant territories, with the social, economic and ecological implications of this farm. The manuscript can be accepted for publication even if a final chapter would be needed with a critical evaluation of the authors on which aspects the supply chain needs to be improved, on which roles donkey breeding must be more considered taking into account the peculiarities. Chapter 13 summarily describes the new markets but the authors, as researchers, have the duty to express a scientific and professional assessment of the development of this supply chain today

Author Response

Reviewer comments

Author responses

The article is interesting and presents a general overview of the donkey supply chain in the relevant territories, with the social, economic and ecological implications of this farm. The manuscript can be accepted for publication even if a final chapter would be needed with a critical evaluation of the authors on which aspects the supply chain needs to be improved, on which roles donkey breeding must be more considered taking into account the peculiarities

Thank you

Final chapter conclusion- included the importance of breeding, animal welfare and sustainable production

Chapter 13 summarily describes the new markets but the authors, as researchers, have the duty to express a scientific and professional assessment of the development of this supply chain today

Thank you. Added the insights of emerging market for donkey milk and its implications for welfare of donkeys and economy.

Reviewer 4 Report

Line 77… a decline of 61.23% (Figures 1&2; Table 1). …

In Figure 2 the data from state number 9 (Himachal Pradesh) have not been included (see Table 1). Why hasn’t this been done? Is this a mistake?

Line 179… Ayo-Odongo et al., in one of their studies [31], based in…

Line 273 What does LMICs mean? This is not indicated by the authors.

Line 414… 14. Conclusions (no number 5).

Finally, mention that the reference section is highly structured, but some references do not scrupulously comply with the ACS reference style format. Review this.

Author Response

Reviewer comments

Author responses

Line 77… a decline of 61.23% (Figures 1&2; Table 1).

Done

In Figure 2 the data from state number 9 (Himachal Pradesh) have not been included (see Table 1). Why hasn’t this been done? Is this a mistake?

Found figure and table 1 are having same information so figure 2 is removed

Line 179… Ayo-Odongo et al., in one of their studies [31], based in…

done

Line 273 What does LMICs mean? This is not indicated by the authors.

done

Line 414… 14. Conclusions (no number 5).

done

Finally, mention that the reference section is highly structured, but some references do not scrupulously comply with the ACS reference style format. Review this

done

Round 2

Reviewer 2 Report

I am satisfied with how the authors have implemented the article with my suggestions and revisions. The article can be accepted.